# The Role of Fibroblast Activation Protein Inhibitor Positron Emission Tomography in Inflammatory and Infectious Diseases: An Updated Systematic Review

**DOI:** 10.3390/ph17060716

**Published:** 2024-05-31

**Authors:** Domenico Albano, Alessio Rizzo, Riemer H. J. A. Slart, Søren Hess, Edel Noriega-Álvarez, Cristina Gamila Wakfie-Corieh, Lucia Leccisotti, Andor W. J. M. Glaudemans, Olivier Gheysens, Giorgio Treglia

**Affiliations:** 1Nuclear Medicine, ASST Spedali Civili Brescia, 25128 Brescia, Italy; domenico.albano@unibs.it; 2Nuclear Medicine Department, University of Brescia, 25121 Brescia, Italy; 3Nuclear Medicine Division, Candiolo Cancer Institute, FPO-IRCCS, 10060 Turin, Italy; alessio.rizzo@ircc.it; 4Department of Nuclear Medicine and Molecular Imaging, University Medical Center Groningen, University of Groningen, 9700 RB Groningen, The Netherlands; r.h.j.a.slart@umcg.nl (R.H.J.A.S.); a.w.j.m.glaudemans@umcg.nl (A.W.J.M.G.); 5Biomedical Photonic Imaging Group, Faculty of Science and Technology, University of Twente, 7522 NB Enschede, The Netherlands; 6Department of Nuclear Medicine, Odense University Hospital, 5000 Odense, Denmark; soren.hess@rsyd.dk; 7Department of Clinical Research, Faculty of Health Sciences, University of Southern Denmark, 5230 Odense, Denmark; 8Department of Nuclear Medicine, University Hospital of Guadalajara, 19002 Guadalajara, Spain; edelnoriega@gmail.com; 9Department of Nuclear Medicine, Hospital Clínico San Carlos, 28040 Madrid, Spain; cristinagwc@gmail.com; 10Instituto de Investigación Sanitaria del Hospital Clínico San Carlos (IdISSC), 28040 Madrid, Spain; 11Section of Nuclear Medicine, Department of Radiological Sciences and Haematology, Università Cattolica del Sacro Cuore, 00168 Rome, Italy; lucia.leccisotti@policlinicogemelli.it; 12Unit of Nuclear Medicine, Fondazione Policlinico Universitario A. Gemelli IRCCS, 00168 Rome, Italy; 13Department of Nuclear Medicine, Cliniques Universitaires Saint-Luc and Institute of Clinical and Experimental Research (IREC), Université Catholique de Louvain, 1200 Brussels, Belgium; olivier.gheysens@saintluc.uclouvain.be; 14Division of Nuclear Medicine, Imaging Institute of Southern Switzerland, Ente Ospedaliero Cantonale, 6500 Bellinzona, Switzerland; 15Faculty of Biomedical Sciences, Università della Svizzera italiana, 6900 Lugano, Switzerland; 16Faculty of Biology and Medicine, University of Lausanne, 1011 Lausanne, Switzerland

**Keywords:** FAPI, PET, inflammation, infection, nuclear medicine, hybrid imaging

## Abstract

The role of fibroblast activation protein inhibitor (FAPI) positron emission tomography/computed tomography (PET/CT) is emerging for the assessment of non-oncological diseases, such as inflammatory and infectious diseases, even if the evidence in the literature is still in its initial phases. We conducted a systematic search of Scopus, PubMed/MEDLINE, Embase, and Cochrane library databases for studies published before 31 December 2023 reporting infectious and inflammatory disease imaging with FAPI PET/CT. We included twenty-one studies for a total of 1046 patients. The most frequent disease studied was lung interstitial disease, investigated in six studies for a total of 200 patients, followed by bone and joint diseases in two studies and 185 patients, IgG4-related disease in 53 patients, and Crohn’s disease in 30 patients. Despite the heterogeneity of studies in terms of study design and technical features, FAPI PET/CT showed a high detection rate and diagnostic role. Moreover, when compared with 2-[^18^F]FDG PET/CT (*n* = 7 studies), FAPI PET/CT seems to have better diagnostic performances. The presence of chronic inflammation and tissue remodeling, typical of immune-mediated inflammatory conditions, may be the underlying mechanism of FAPI uptake.

## 1. Introduction

Positron emission tomography/computed tomography (PET/CT) is a hybrid imaging tool with an increasing range of uses and applications. The most frequent radiopharmaceutical utilized to perform such a diagnostic examination is 2-deoxy-2-[^18^F]-fluoro-*D*-glucose (2-[^18^F]FDG), which can reveal lesions with increased metabolism based on their increased glycolytic activity. In addition to applications in oncology, 2-[^18^F]FDG PET has demonstrated a high impact in the evaluation of infectious and inflammatory diseases, as well [1,2]. With the aim of investigating metabolic pathways as alternatives to glucose metabolism, various PET radiopharmaceuticals have been projected as alternatives to 2-[^18^F]FDG. In this setting, fibroblast activation protein (FAP) inhibitors (FAPI), labeled with either ^18^F or ^68^Ga, are emerging as promising radiopharmaceuticals. FAP is a type II transmembrane serine protease belonging to the dipeptidyl peptidase 4 family, and it is highly over-expressed on the membrane of cancer-associated fibroblasts (CAFs) in about 90% of epithelial-derived cancers [3], but it may also be present in the presence of tissue damage, remodeling, or chronic inflammation, and therefore in benign conditions [4]. In contrast, healthy tissues have low expression of FAP [5]. The initial FAPI variants (called FAPI-01 and FAPI-02) were developed in 2018 and were labeled with ^68^Ga, showing favorable tracer retention in pathological tissues and low background uptake in normal healthy tissues, providing high-contrast images and an optimal target-to-background uptake ratio (TBR). Then, FAPI-04 was recognized as the most optimal radiotracer out of all modified FAPIs for PET imaging studies, demonstrating a higher tumor uptake in murine xenograft models than FAPI-02 [6] and also in humans with different cancers. In the oncological field, the effective usefulness of FAPI PET/CT has been evaluated in some research and preliminary evidence has shown positive results for some kinds of tumors, also compared to 2-[^18^F]FDG PET/CT [3,7,8,9]. Similar to that of 2-[^18^F]FDG, the uptake of radiolabeled FAPI has also been described in several benign findings [10], which underlines similar specificity issues of FAPI uptake for oncological lesions. Thus, some more recently published clinical trials anticipated a function of FAPI PET for the evaluation of inflammatory and infectious diseases due to the radiolabeled FAPI uptake as a consequence of fibroblast activation in tissue remodeling, e.g., the application of FAPI PET in rheumatic diseases [4]. These recent data on the role of FAPI PET/CT in infectious and inflammatory conditions are receiving emerging attention from the scientific community. The desirable increased availability of FAPI radiotracers worldwide will surely increase studies on these topics, with a final goal of FAPI PET/CT use in clinical practice for selected infectious and inflammatory conditions in the coming years. For these reasons, our systematic review aims to recap the main data about the actual clinical potential usefulness of FAPI PET/CT radiopharmaceuticals in infection and inflammation.

## 2. Materials and Methods

### 2.1. Protocol

We conducted this systematic review following a preset protocol, and we used the “Preferred Reporting Items for a Systematic Review and Meta-Analysis” (PRISMA 2020 statement) as a guideline for development and reporting. The complete PRISMA checklist is available in Appendix A. We started formulating this review query: What is the diagnostic role of FAPI PET in infectious and inflammatory diseases? Following the Population, Intervention, Comparator, and Outcomes (PICO) framework, we defined the following criteria for research inclusion: studies performed on patients with suspected infectious or inflammatory disease (Population), having PET with FAPI (Intervention), and with or without a comparison with standard-of-care imaging (Comparator) [11]. Our main outcome was the diagnostic role of FAPI PET in infectious or inflammatory diseases.

### 2.2. Strategy for Literature Research and Information Sources

Starting from the review query, we conducted a literature search of Embase, PubMed/MEDLINE, Scopus, and Cochrane library databases with the aim of extracting any significant published studies based on the potential role of FAPI PET in infectious or inflammatory diseases. The ClinicalTrials.gov database was additionally analyzed for ongoing studies (access date: 31 December 2023). The search algorithm used for this research was composed of the combination of these terms: (A) “PET” OR “positron” AND (B) “FAP” or “FAPI” AND (C) “infection” OR “infectious” OR “infective” OR “inflammation” OR “inflammatory”. No restrictions regarding the year of release were applied. Regarding the language, only papers written in the English language were considered. To enlarge our research, references of the retrieved articles were also screened to search for additional records. The last update to the literature inquiry was set on 31 December 2023.

### 2.3. Eligibility Criteria

All the original research describing the potential role of FAPI PET in infectious and inflammatory diseases was included in this review. We excluded the non-original studies, like editorials, comments, meta-analyses, reviews, and letters concerning the selected topic, as well as original articles not in the field of interest (including also preclinical studies). Moreover, articles based upon fewer than 5 patients or case reports were excluded from the systematic review. Two researchers (D.A. and A.R.) independently reviewed the titles and abstracts of the records, and independently reviewed the full-text version of the articles to evaluate their suitability. In the case of a disagreement, a third opinion (G.T.) was involved in a consensus meeting to settle any discrepancy.

### 2.4. Process of Data Collection and Data Extraction

Two researchers (D.A. and A.R.) read all the studies separately and analyzed the information in the entire manuscript and all tables, and images. The findings extrapolated from any piece of research were as follows: general article information (authors, nationality, year of publication, number of included subjects, methodology); disease details; technical variables (PET device used, kind of tracer, kind of hybrid imaging procedure, injected activity, uptake time between FAPI injection and image scan), and the main findings.

### 2.5. Quality Assessment (Risk of Bias Assessment)

For the investigation of the risk of bias in each study and for the analysis of the relevance of each paper to the review query, we referred to QUADAS-2, an instrument created to assess the quality of studies based on the accuracy of diagnostic procedures. The quality of the included studies in the systematic review was analyzed by two researchers independently. They described four different domains (patient selection, index test, reference standard, and flow and timing) for the assessment of the bias risk, and three domains (patient selection, index test, and reference standard) for applicability.

### 2.6. Statistical Analysis

Meta-analysis (quantitative synthesis) was not performed as significant heterogeneity among the selected studies (such as the different kinds of diseases, different endpoints investigated) was expected.

## 3. Results

### 3.1. Literature Search and Selection of Studies

After the comprehensive literature search, we obtained 211 items. Among them, 190 records were excluded using the previous eligibility criteria (66 because of duplicate records, 30 because of the field of interest, 18 because of reviews/editorials, 72 as case series or case reports, and 4 as preclinical studies). Finally, 21 articles were considered worthy of inclusion in the systematic review (qualitative synthesis) [12,13,14,15,16,17,18,19,20,21,22,23,24,25,26,27,28,29,30,31,32]. No additional records were added after screening the references of the eligible studies.

In Figure 1, we summarized the flow chart of the study selection process. The complete list of excluded studies (with reasons for exclusion) is present in the Appendix A.

### 3.2. Study Characteristics

The 21 studies fulfilling the criteria for inclusion in the qualitative analysis of this systematic review included a total of 1046 patients suspected of different types of infectious and inflammatory diseases. A more detailed analysis of these studies is available in Table 1 and Table 2. Regarding general study data (Table 1), the included studies were published between 2020 and 2023 in Asia and Europe. Most research were performed in China (*n* = 12), followed by Germany (*n* = 6) and Austria (*n* = 2). Thirteen studies were conducted with a retrospective design [12,14,15,16,17,18,21,22,23,26,27,30,31], whereas the remaining eight were prospective. All the studies included in this systematic review were monocentric. The number of recruited patients in any study ranged from 6 to 182, with an average/median age range from 23 to 71 years, and the rate of men varied from 25% to 83%. Among the included studies, the performance of FAPI PET/CT was explored in different infectious and inflammatory diseases. Lung infectious/inflammatory diseases were the most common conditions investigated in six papers [12,14,16,23,24,26] and 200 patients; bone and joint diseases in two papers [18,22], for a total of 185 patients; IgG4-Related disease (IgG4-RD) in two papers [29,32], for a total of 53 patients; and Crohn’s disease in two papers [15,20], for a total of 30 patients. In three articles [13,21,30], several mixed benign conditions, including infectious and inflammatory diseases, were also investigated. In the remaining articles, single diseases were studied, such as thyroiditis [28], myocarditis [27], systemic sclerosis [17], rheumatoid arthritis [19], light-chain cardiac amyloidosis [25], and renal fibrosis [31]. In seven studies [13,14,19,22,24,29,32], FAPI PET/CT performances were compared with 2-[^18^F]FDG PET/CT. As shown in Table 2, the index test features were very heterogeneous. Most studies administrated [^68^Ga]Ga-DOTA-FAPI-04 [13,15,16,17,18,19,22,25,26,28,29,30,31,32], two studies [^68^Ga]Ga-DOTA-FAPI-46 [23,24], one study [^18^F]AlF-NOTA-FAPI-04 [14], one study [^68^Ga]Ga-DATA5m.SA.FAPI [20], and one study [^18^F]-FAPI-74 [12]; one study used both [^68^Ga]Ga-DOTA-FAPI-04 and [^68^Ga]Ga-DOTA-FAPI-02 [21]; and one paper did not note the radiopharmaceutical form of radiolabeled FAPI in the text [27]. The radiotracer activity injected was very heterogeneous. When evaluated using absolute values, the administered activity ranged from 55 to 336 MBq, and when reported using relative values, it ranged from 1.5 to 4.81 MBq/kg. In addition, there were large differences in uptake period, with time spans (from 10 to 180 min) between radiotracer injection and PET scanning. Only one study [23] performed a dynamic scan, and only for a small number of patients recruited. Seventeen papers had PET/CT as a hybrid imaging method [12,13,14,15,16,17,18,19,22,23,24,25,26,27,28,29,30,31,32], one study used a PET/MRI scanner [20], and one employed both PET/CT and PET/MRI devices [22]. Qualitative (visual) and semiquantitative analyses were performed in all the articles included in this systematic review. The most common semiquantitative parameters described in the studies were the mean and maximum standardized uptake values (SUVmean and SUVmax, respectively) and TBR. Less frequent semiquantitative parameters investigated were metabolic active volume (MAV), molecular volume (MV), total lesion FAPI (TL-FAPI), and fibrotic active volume (FAV).

### 3.3. Results of Individual Studies (Qualitative Synthesis)

#### 3.3.1. Inflammatory Lung Diseases

Concerning inflammatory/infectious lung diseases, in the first study, Röhrich et al. [23] used FAPI PET/CT in 15 patients affected by fibrotic interstitial lung disease (fILD) and concomitant suspected lung cancer (LC) as a complication. FAPI PET/CT results were acquired at different time points (10, 60, and 180 min after the radiotracer administration); in three cases, a dynamic study over 40 min was performed instead of an early scan. Both conditions (fILD and LC lesions) displayed elevated radiopharmaceutical uptake, with density-corrected SUVmax and SUVmean slightly higher in LC cases (16.7 and 6.4, respectively) than fILD (11.2 and 4.29, respectively). TBRs were relatively stable in fILD, while showing a tendency to increase over time in LC. Despite being performed in only three patients, dynamic PET revealed a significant difference in the time-activity curves for LC and fILD, i.e., fILD lesions displayed an early peak and a subsequent slow reduction in signal intensity, whereas LC had a late peak, followed by a gradual washout phase. These preliminary data indicated FAPI PET/CT as a promising new imaging modality for the study of fILD and suspected LC, with dynamic imaging being the most promising for direct differentiation between benign and malignant lesions. The authors also stressed that further evaluation is necessary to establish whether FAPI holds any real added value over standard-of-care CT and 2-[^18^F]FDG, e.g., to differentiate inactive from activated fibrosis and in turn predict treatment responses with antifibrotic drugs where 2-[^18^F]FDG has shortcomings.

Bergmann et al. [26] actually addressed the question of patients’ outcomes and treatment monitoring in their single-center pilot study. Based on 21 patients and 21 controls, they studied the role of FAPI PET/CT in systemic sclerosis-associated ILD, the most common fatal complication in systemic sclerosis. Patients with systemic sclerosis-associated ILD had a significantly higher FAPI uptake than controls. Considering only ILD patients, FAPI uptake was higher in cases of extensive disease compared to limited disease, in cases of previous ILD progression compared to stable disease, or in cases with high EUSTAR activity scores compared to low EUSTAR scores. Moreover, baseline FAPI uptake was associated with the risk of progression independent of the extent of involvement on high resolution CT scans and the forced vital capacity at baseline. Finally, when monitoring treatment responses, changes in FAPI uptake on sequential scans were concordant with radiological and clinical responses in patients treated with nintedanib. Thus, the authors concluded that FAPI PET has the potential to improve risk assessments of patients with systemic sclerosis-associated ILD, allowing for the earlier treatment and dynamic monitoring of molecular treatment responses.

Yang et al. [16] investigated in vitro and in vivo whether the expression intensity of FAP could be used as a potential readout to estimate or measure the amounts of activated fibroblasts in ILD. They demonstrated that FAP expression was upregulated significantly in the prodromal phases of interstitial diseases characterized by lung fibroblast activation which are expressed by collagen cells. Subsequent immunohistochemical analysis validated FAP expression as nearly associated with the abundance of fibroblastic foci on human lung biopsy. This evidence demonstrated a confirmation in PET/CT studies where the uptake expressed as SUV was significantly correlated to a pulmonary function decrease in patients with ILD [16].

Another study [14] confirmed that inflammatory lung conditions might have an abnormal FAPI uptake, and aimed to compare the uptake of FAPI and 2-[^18^F]FDG in both malignant and inflammatory lesions to assess their value for differential diagnoses. Both tracers exhibited significantly lower uptake in inflammatory lesions compared to malignant lung lesions, with statistically significant differences in terms of SUVmax, SUVmean, and TBR (*p* < 0.001). Among the inflammatory findings, those with the highest uptake were infected bronchiectasis, followed by post-obstructive pneumonia and pneumonia.

In the most recent study, eight patients with idiopathic pulmonary fibrosis underwent FAPI PET and high-resolution CT and were compared with a control group without any fibrosing pulmonary disease [12]. Fibrotic changes in pulmonary parenchyma showed a markedly increased FAPI uptake in both the visual and semiquantitative analyses. Moreover, FAPI uptake was significantly correlated with lung density expressed by Hounsfield units and clinical parameters (forced vital capacity).

Instead, Sviridenko et al. [24] analyzed the role of FAPI PET in post-COVID-19 patients with persistent lung abnormalities and compared its diagnostic capacity to 2-[^18^F]FDG PET/CT. FAPI PET was positive in all six patients included, while 2-[^18^F]FDG PET/CT scans were negative. Both FAPI and 2-[^18^F]FDG PET scans were negative in controls without COVID-19. These findings suggest that the structural lung changes in post-COVID-19 patients are triggered by fibrotic repair processes and not by underlying inflammation, and underline the potential usefulness of FAPI PET in monitoring long-COVID lung alterations.

#### 3.3.2. Bone and Joint Diseases

In patients with suspected bone and joint diseases, two studies were available [18,22]. Wang et al. [18] included 103 patients with symptomatic hip arthroplasty suspected for infection; 28 patients were positive for infection, and they demonstrated that FAPI PET may be useful in differentiating periprosthetic hip joint infection and aseptic failure, with the most optimal SUVmax threshold derived from receiver operating curve analysis being 7.53 (sensitivity 100% and specificity 72%). Better results were obtained by analyzing uptake patterns similar to those used in 2-[^18^F]FDG imaging, and this yielded sensitivity and specificity values of 100% and 93%. Furthermore, as the only study included, they analyzed radiomics features and derived significant differences between infection and aseptic loosening. Qin et al. [22] compared the diagnostic role of FAPI PET in distinguishing bone metastases from benign diseases (including osteofibrous dysplasia, periodontitis, degenerative bone diseases, arthritis, and other inflammatory or trauma-related abnormalities). Although the uptake of FAPI was often higher in bone metastases, this finding was not able to discriminate between benign and malignant lesions. In this setting, FAPI PET showed the potential to locate and evaluate the extent of both malignant tumor and benign diseases in bones and joints.

#### 3.3.3. IgG4-Related Diseases

Two studies [29,32] focused on IgG4-RD and were concordant about the positive impact of FAPI PET in the detection and differential diagnosis of IgG4-RD.

In the first study [29], FAPI PET/CT demonstrated a higher positive rate than 2-[^18^F]FDG PET/CT in detecting involvement, especially in the pancreas, bile duct/liver, and salivary and lacrimal glands. FAPI may be especially advantageous in the head-and-neck region due to the low background uptake because the physiologic 2-[^18^F]FDG uptake in salivary glands and extraocular muscles may mask the disease. FAPI found additional organ affection in about 50% of patients recruited. However, all 2-[^18^F]FDG-avid lymph nodes were FAPI-negative, likely because IgG4-RD lymphadenopathies usually lack the storiform fibrosis that otherwise characterizes IgG4-RD. In the second study [32], the authors showed that FAPI and 2-[^18^F]FDG could be used to study different stages of IgG4-RD as a fibro-inflammatory disease, particularly in discriminating between inflammatory (FAPI-negative and 2-[^18^F]FDG-positive) and not inflammatory fibrosis (FAPI-positive and 2-[^18^F]FDG-negative). FAPI uptake did not fully correlate with 2-[^18^F]FDG uptake, suggesting that inflammation and fibrosis are not fully interconnected. They also found that whereas 2-[^18^F]FDG uptake was very responsive to anti-inflammatory therapy, FAPI was not. These findings open up the possibility of better personalizing the management and treatment choices of these patients, like the choice of therapy (anti-fibrotic and/or anti-inflammatory). However, more solid data need to be shared to confirm or counter these results in this disease or other fields.

#### 3.3.4. Crohn’s Disease

Regarding Crohn’s disease, two papers were available [15,20]. In the first one [15], FAPI PET/CT was shown to be a promising tool for assessing the activity of patients with Crohn’s disease, particularly identifying 42 of 45 segmental lesions detected by endoscopy with a diagnostic accuracy higher than computed tomography enterography; i.e., the overall sensitivity and specificity of 93% and 72% versus 87% and 44%, respectively. FAPI uptake was proportional to the severity of the endoscopic lesions, and FAPI was more sensitive than CTE for detecting severe lesions, but at the same time the FAPI detection rate for mild-to-moderate lesions was also higher. Furthermore, FAPI uptake was significantly associated with some samples and clinical parameters, such as fecal calprotectin, C-reactive protein, Crohn’s disease endoscopy index of severity, and the simple endoscopic score for Crohn’s disease. It is worth noting that results in reality may be even better for FAPI; first, the colonoscopy was incomplete in 25% of patients and FAPI may be especially advantageous in these cases. Second, some segments were FAPI- and CTE-positive even without endoscopic lesions that may represent submucosal fibrosis. The other study [20] investigated the potential role of FAPI PET/MRI enterography for differentiating active intestinal inflammation and fibrosis. FAPI uptake was significantly associated with the presence of fibrotic tissue and was lower in inflammatory lesions than in fibrotic ones. The differentiation between active intestinal inflammation and fibrosis has implications for treatment, and it is extremely challenging with currently available standard examinations. In this clinical scenario, FAPI may reveal new information and be helpful in treatment decision algorithms.

#### 3.3.5. Other

In three articles [13,21,30] mixed benign lesions were analyzed; the main evidence of these studies was a higher uptake of FAPI than 2-[^18^F]FDG in infectious diseases and fibrotic conditions, while for inflammatory diseases a similar detection rate was observed and the performance of FAPI was generally considered the best in fibrotic inflammatory disorders. Despite significant uptake, benign lesions showed a lower uptake than malignant conditions. In the study by Deng et al. [13], FAPI was also compared to conventional bone scintigraphy in patients with benign bone disorders and was found to be inferior.

Other studies [17,19,25,27,28,30,31] were mono-thematic (myocarditis, thyroiditis, renal fibrosis, light-chain cardiac amyloidosis, rheumatoid arthritis, systemic sclerosis) and showed the potential clinical usefulness of FAPI PET/CT in all these settings.

### 3.4. Quality (Risk of Bias) Assessment

Figure 2 displays the risk of bias and concerns regarding the applicability for papers analyzed. Regarding risk of bias and applicability concerns, in most studies these risks are low (reported in green color in the figure).

## 4. Discussion

The role of FAPI PET is developing for the assessment of non-oncological conditions, such as inflammatory and infectious diseases, even if the evidence in the literature is still in its early stages [33,34]. As stated in the recent literature, focal radiolabeled FAPI uptake is related to active tissue remodeling in patients with immune-mediated inflammatory diseases, which are characterized by the presence of chronic inflammation and tissue response. Consequently, benign lesions may be frequently characterized by the presence of increased FAPI uptake. Interestingly, the same infectious/inflammatory lesions are not always positive to 2-[^18^F]FDG PET, indicating that inflammation and the related tissue response could be disentangled by the combination of both radiopharmaceuticals [32]. Concerning 2-[^18^F]FDG [1,2], the mechanism of uptake seems to be related to the increased metabolism of some inflammatory cells, such as macrophages, granulocytes, and lymphocytes, resulting in a sharply increased demand for glucose and the expression of glucose transporters. Conversely, for FAPI tracers the mechanism is completely different: FAP is a membrane-anchored glycoprotein belonging to the dipeptidyl peptidase family. FAP characteristically possesses both post-proline peptidase and endopeptidase activity, enabling FAP to cleave proteins in the surrounding tissue, thus promoting protein degradation and matrix remodeling. FAP is overexpressed on CAFs’ cell membrane and stroma and can be an indirect expression of extracellular matrix remodeling, typical, for example, in several inflammatory and infectious conditions. FAP is also overexpressed in synovial activated fibroblasts (SAF) that secrete TGF-β. In particular, in tissue remodeling sites FAP is overexpressed by cells like fibroblasts and chondrocytes [8,35]. Fibroblasts are abundant in the human body and participate in various inflammatory, infectious, and immune responses. When cells and tissues suffer abnormal conditions, such as oxidative stress, FAP is expressed; conversely, it undergoes downregulation when the internal environment is stable. For example, in periprosthetic infection cases, bacteria involved in the infective disease can synthesize a biofilm, an extracellular matrix, which might stimulate the expression of FAP [36]. Another function of FAP is to destroy and remodel the extracellular matrix with the help of endopeptidase action and then to stimulate the development of microvessels and complexes with other proteins to conduct intercellular signals. Moreover, the relationship between FAP and macrophages, which are typically inflammatory cells, is well known. FAP may stimulate the activation of macrophages in different settings. At the same time, macrophages may express FAP receptors on their cell surface [37]. On the other hand, other authors have proposed a different mechanism of uptake related to the increased vascularity and capillary permeability due to inflammatory response, resulting in high perfusion and blood-pool effects [38].

FAPI tracers are characterized by several benefits over 2-[^18^F]FDG, such as higher TBR, fast renal clearance, and tracer kinetics, resulting in a shorter time interval between injection and imaging and no dependency on blood glucose and resting. From a theoretical point of view, these advantages may make it easier to perform a FAPI PET scan, but a prerequisite for introducing a novel tracer is comparable diagnostic performance. Confirming this point, all studies in the literature directly comparing FAPI and 2-[^18^F]FDG in the infection and inflammation field showed a potential added value of FAPI over 2-[^18^F]FDG or a complementary role.

In fact, it seems that diseases characterized by fibrotic tissues, infectious conditions and benign lesions presented higher uptake and detection rates with FAPI than 2-[^18^F]FDG, whereas diseases with a predominant inflammatory component showed an equivalent or lower detection efficacy of FAPI to 2-[^18^F]FDG [13,21,32]. The study by Schmidkonz et al. [32] is a perfect example: the combination of FAPI and 2-[^18^F]FDG PET/CT findings helped to discriminate between inflammatory and not inflammatory IgG4-RD with good accuracy. The 2-[^18^F]FDG PET-positive lesions demonstrated a high lymphoplasmacytic infiltration of IgG4 cells in histological analysis, whereas FAPI PET-positive lesions demonstrated a high density of activated fibroblasts expressing FAP.

These results underlined the possibility of better stratifying patients with inflammatory/infectious diseases with a combination of both PET studies.

Other studies [19,24,29] described the superior diagnostic capacity of FAPI over 2-[^18^F]FDG for the study of several infectious and inflammatory conditions, like ILD, rheumatoid arthritis, and IgG4-RD.

The findings derived from the literature include the fact that FAPI uptake is more specific for fibrotic than inflammatory disorders, but the temporal and spatial relationships between these two mechanism in the parthenogenesis of several diseases are still unclear.

With the use of FAPI tracers for in vivo imaging, we could try to answer this question and elucidate the disease progression and underlying mechanisms phases.

However, not all infectious and inflammatory conditions were investigated with FAPI and the evidence that we have now is less solid and robust compared to that on 2-[^18^F]FDG. Moreover, a direct comparison is not so easy due to the different stages of diseases included. We can probably suppose a complementary role of FAPI and 2-[^18^F]FDG, with the possibility of studying several infectious and inflammatory diseases from different points of view and providing different information.

An interesting point in favor of FAPI was the excellent detection rate in long-COVID patients with lung alterations [24]. In the literature, incidental COVID-19 interstitial pneumonia-related conditions were reported to be detected by 2-[^18^F]FDG PET [39,40,41], but the effective mechanism of this uptake is not yet clear. It seems that acute-phase pneumonia and aggressive disease, which often present with characteristic CT findings such as ground-glass opacities and consolidation, are more probable than 2-[^18^F]FDG-avid. Instead, Sviridenko et al. [24] reported six patients with 2-[^18^F]FDG negative COVID-19 pneumonia but positive at FAPI PET, suggesting a superiority of this novel radiotracer in late-disease, chronic pneumonia or long-term complications such as pulmonary fibrosis.

Another finding that benefits the regular employment of FAPI over 2-[18]FDG is the absence or low physiological accumulation into the bone marrow and brain, resulting in a higher TBR and perhaps detection rate in bone and brain diseases.

These properties make FAPI a hopeful and clinically practicable biomarker to quantitatively document ongoing fibrosis because it is a membrane-anchored protein in active fibroblasts.

To date, FAPI PET has not been compared with other nuclear medicine radiopharmaceuticals, except for 2-[^18^F]FDG. In particular, no comparison has been made with radiolabeled white blood cell scintigraphy (WBCS), which is now reasoned to be the best nuclear medicine imaging modality in specific settings, such as in discriminating between periprosthetic joint infection and aseptic loosening. However, from a theoretical point of view, these two imaging tools (FAPI PET and WBCS) seem to be more complementary than alternative. FAPI seems to be more “avid” in inflammatory/fibrotic conditions than in acute infections, which is usually a peculiarity of radiolabeled WBC.

The articles included in this review were very heterogeneous from a methodological point of view; for example, the papers employed different radiopharmaceutical forms and uptake times, and the reported radiotracer activity injected varied significantly among the single studies. Of course, this is due to the recent development of this radiotracer and the consequent absence of international operative guidelines and shared consensus. Moreover, the emerging body of literature concerning the employment of this tracer encompasses different oncological and non-oncological diseases with peculiar characteristics, further complicating the search for a shared protocol even more. This makes it difficult to compare the included studies ideally.

The application of dynamic PET, especially with “new” scanners, may help to study patients and discriminate more easily between infection and inflammation.

Despite these biases, the preliminary results about the usefulness of FAPI PET in infectious and inflammatory diseases are auspicious. In almost all cases, the detection rate and overall diagnostic role described are excellent and better than conventional imaging tools. The results presented in this systematic review suggest a very promising clinical impact in different conditions. The strongest evidence, for the number of patients enrolled and the number of studies available, is available in interstitial lung disease, Crohn’s disease, IgG4-RD, and bone and joint diseases. Nevertheless, it is fundamental to obtain more solid data and to design appropriate clinical trials to provide a clearer statement of FAPI’s real usefulness in this field.

Another peculiarity of FAPI that differentiates it from other radiotracers, such as 2-[18]FDG, is the potential theragnostic application. FAPI may be radiolabeled with therapeutic nuclides such as ^90^Y, ^177^Lu, ^225^Ac, and ^153^Sm for therapeutic purposes. Preliminary reports about the theragnostic role of FAPI have been reported in advanced sarcoma, pancreatic adenocarcinoma, and breast cancer [42]. Even in non-oncological fields, the theragnostic application of FAPI may be supposed. For example, in cardiovascular diseases, novel therapies based on tailored agents such as small molecules, RNAs, or cell products have been emerging as promising, but they will require personalized guidance to determine the right time at which the target mechanism is activated pathologically in the target tissue, but is not physiologically or reactively active in other tissues/organs. Even more recent treatments, like antifibrotic therapies (including FAP-directed chimeric antigen receptor-expressing T-cells) have been emerging, and FAPI-PET may assist in identifying appropriate subjects who will be able to benefit, by providing information about the heart and remote organ uptake.

However, it is mandatory to generate more solid data and well-designed and multi-center studies and clinical trials to increase the comprehension of FAPI’s real usefulness in the theragnostic area.

## 5. Conclusions

In conclusion, radiolabeled FAPI PET tracers are promising novel agents with peculiar features and clear advantages over 2-[^18^F]FDG. Potentially, FAP-targeting tracers may substitute or be complementary to the current standard-of-care imaging in several clinical scenarios, including inflammatory and infectious diseases.

## Figures and Tables

**Figure 1 pharmaceuticals-17-00716-f001:**
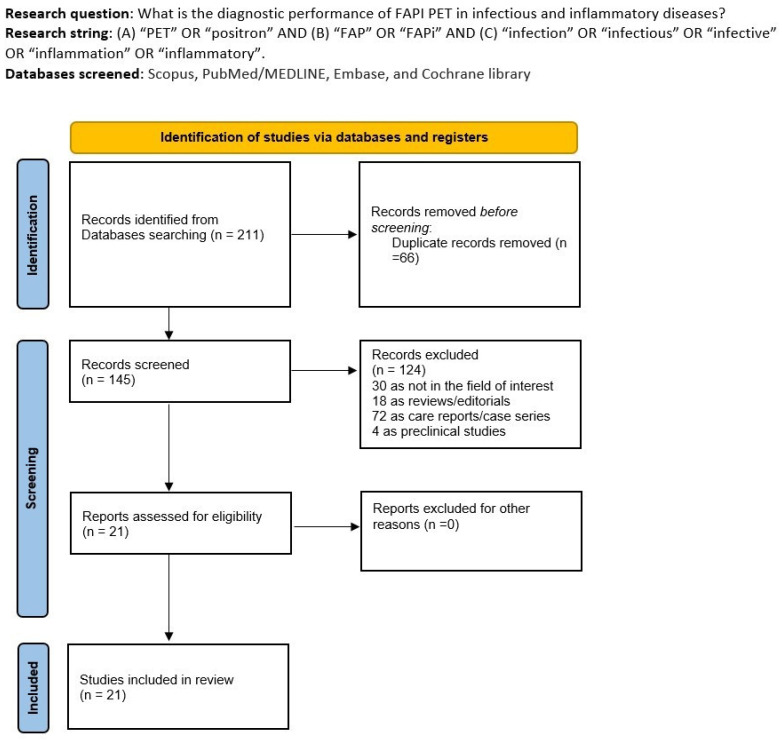
Comprehensive summary of the study selection process.

**Figure 2 pharmaceuticals-17-00716-f002:**
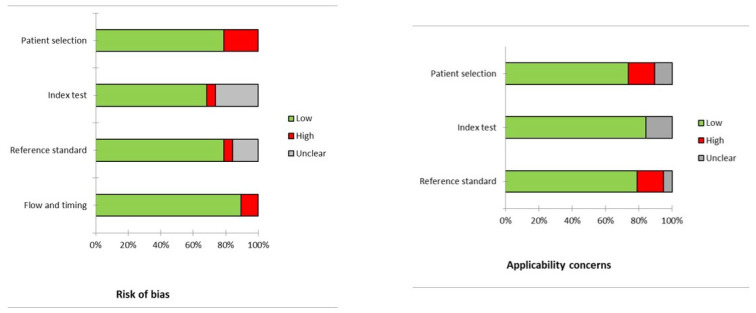
Quality assessment according to QUADAS-2. In the vertical axis, the domains regarding the risk of bias or applicability concerns are illustrated. In the horizontal axis, the percentage of studies is represented. Most of the studies have low risk of bias or applicability concerns (in green), and only a few studies showed a high risk of bias or applicability concerns (in red).

**Table 1 pharmaceuticals-17-00716-t001:** General study information, patients’ key characteristics, and main findings.

First Author, Ref.	Country	Year	Type of Study	Disease	No. of Patients	Mean/Median Age (Years)	Gender Male (%)	Comparison with 2-[^18^F]FDG	Main Findings
Mori Y [12]	Chile	2023	retrospective	Idiopathic pulmonary fibrosis	8	Median: 71	50%	no	FAPI uptake was higher in fibrotic areas and correlated with HU and reduced vital capacity.
Li Y [13]	China	2023	prospective	Several benign diseases (inflammatory disease, infectious disease, benign tumors, fibrotic disease, benign bone disease)	111	Mean: 53	48%	yes	FAPI PET demonstrated a better detection rate and increased uptake in fibrotic tissues, infectious diseases, and benign tumors than 2-[^18^F]FDG. In inflammatory disease, an equivalent detection efficacy between FAPI and 2-[^18^F]FDG was demonstrated. In non-malignant bone diseases, FAPI PET had a lower uptake and a comparable detection rate compared to [^99m^Tc]MDP.
Qiao K [14]	China	2023	retrospective	Various lung lesions	67	Mean: 63.39	81%	yes	FAPI PET may differentiate between inflammatory and neoplastic lung diseases.
Chen L [15]	China	2023	retrospective	Crohn’s disease	16	Median: 23	69%	no	FAPI PET/CT may be useful for detecting the status of Crohn’s disease.
Yang P [16]	China	2023	retrospective	Interstitial lung diseases	83	Nr	56%	no	FAPI PET/CT may describe the profibrotic activity of interstitial lung diseases. FAPI uptake was directly associated with pulmonary lung function decrease.
Treutlein C [17]	Germany	2023	retrospective	Systemic sclerosis	14	Median: 57	50%	no	FAPI PET/CT may be a valid alternative to study cardiac fibroblast activity through being able to detect myocardial fibrosis related to systemic sclerosis.
Wang Y [18]	China	2023	retrospective	Bone and joint diseases	103	Mean: 60	47%	no	FAPI PET may be helpful in the differential diagnosis of periprosthetic hip joint infection and aseptic failure.
Luo Y [19]	China	2023	prospective	Rheumatoid arthritis	20	Mean: 55	25%	yes	FAPI PET is better than 2-[^18^F]FDG in detecting rheumatoid arthritis.
Scharitzer M [20]	Austria	2023	prospective	Crohn’s disease	14	Mean: 45	71%	no	PET/MR enterography uptake was correlated with histopathologically assessed bowel wall fibrosis in Crohn’s disease
Dabir M [21]	Germany	2023	retrospective	Several benign lesions (pancreatitis/pancreas fibrosis, prostatic fibrosis, lung fibrosis/infection, testicular fibrosis, liver fibrosis, thyroiditis, arteriosclerosis, esophagitis)	155	Median: 67	55%	no	FAPI uptake was significantly lower in benign lesions than malignant lesions.
Qin C [22]	China	2022	retrospective	Bone and joint diseases	82	Mean 56.65	57%	yes	FAPI accumulated in both bone metastases and some benign diseases of the bones and joints. Although the uptake of FAPI was often higher in bone metastases, this finding cannot be used to distinguish between benign and malignant lesions.
Röhrich M [23]	Germany	2022	retrospective	Interstitial lung diseases	15	Mean 71.2	nr	no	FAPI PET/CT may help investigate fibrotic interstitial lung disease
Sviridenko A [24]	Austria	2022	prospective	COVID-19 pneumonia	6	Mean: 63.3	83%	yes	FAPI PET may be useful in evaluating long COVID.
Wang X [25]	China	2022	prospective	Light-chain cardiac amyloidosis	30	Mean: 59.1	67%	no	FAPI PET/CT may help recognize light-chain cardiac amyloidosis by detecting myocardial fibroblast activation correlated with myocardial remodeling.
Bergmann C [26]	Germany	2021	retrospective	Systemic sclerosis-associated interstitial lung disease	21	Nr	nr	no	In patients with systemic sclerosis-associated interstitial lung diseases, FAPI uptake is associated with fibrotic activity and disease progression.
Finke D [27]	Germany	2021	retrospective	Myocarditis	26	Nr	nr	no	FAPI PET/CT may be useful to detect checkpoint inhibitors in myocarditis.
Liu H [28]	China	2021	prospective	Thyroiditis	27	Nr	nr	no	Chronic lymphocytic thyroiditis can cause incidental FAPI uptake in the thyroid.
Luo Y [29]	China	2021	prospective	IgG4-Related disease	26	Mean: 51.5	77%	yes	FAPI PET/CT had a higher detection rate than 2-[^18^F]FDG PET/CT in detecting involvement in the pancreas, bile ducts, liver, and lacrimal glands.
Zheng S [30]	China	2021	retrospective	Several benign lesions (inflammatory lymph nodes, osteoarthritis, periodontitis, TBC, esophagitis, pneumonia, pancreatitis, cirrhosis, mastoiditis, prostatitis, appendicitis, renal amyloidosis)	182	Median: 57	62%	no	Benign lesions may have increased FAPI uptake, but usually less than malignant diseases. However, there is a wide overlap of SUVmax range between benign and malignant lesions.
Zhou Y [31]	China	2021	retrospective	Renal fibrosis	13	Mean: 42	62%	no	FAPI PET/CT has high sensitivity in detecting renal fibrosis.
Schmidkonz C [32]	Germany	2020	prospective	IgG4-related disease	27	Mean: 54.9	70%	yes	FAPI PET/CT allows the discrimination between inflammatory and fibrotic activity in IgG4-related disease, the latter being characterized by increased FAPI uptake.

Nr: not reported; PET: positron emission tomography; CT: computed tomography; MR: magnetic resonance; FAPI: fibroblast activation protein inhibitors.

**Table 2 pharmaceuticals-17-00716-t002:** Key characteristics of index tests in the included studies.

First Author, Ref.	Tracer	Hybrid Imaging	Scanner	Administered Activity	Uptake Time (min)	Image Analysis
Mori Y [12]	[^18^F]FAPI-74	PET/CT	Biograph mCT Flow Siemens	199–239 MBq	60	Qualitative, semiquantitative (SUVmean, SUVmax, FAV)
Li Y [13]	[^68^Ga]Ga-DOTA-FAPI-04	PET/CT	Umi 780 United-Imaging	1.85 MBq/kg	60	Qualitative, semiquantitative (SUVmax)
Qiao K [14]	[^18^F]AlF-NOTA-FAPI-04	PET/CT	GEMINI TF Big Bore, Philips	4.81 MBq/kg	60	Qualitative, semiquantitative (SUVmean, SUVmax, TBR)
Chen L [15]	[^68^Ga]Ga-DOTA-FAPI-04	PET/CT	Umi 780 United-Imaging	1.85–2.96 MBq/kg	60	Qualitative, semiquantitative (SUVmean, SUVmax, TBR)
Yang P [16]	[^68^Ga]Ga-DOTA-FAPI-04	PET/CT	Discovery ST, GE	Na	Na	Qualitative, semiquantitative (SUVmean)
Treutlein C [17]	[^68^Ga]Ga-DOTA-FAPI-04	PET/CT	Biograph mCT 40 Siemens	1.5 MBq/kg	15	Qualitative, semiquantitative (SUVmax, SUVmean, MAV, TBR)
Wang Y [18]	[^68^Ga]Ga-DOTA-FAPI-04	PET/CT	uMI510, United Imaging	1.8–2.4 MBq/kg	Na	Qualitative, semiquantitative (SUVmax, radiomic features)
Luo Y [19]	[^68^Ga]Ga-DOTA-FAPI-04	PET/CT	Biograph 64 Truepoint TrueV, Siemens; Polestar m660, SinoUnion	104.3 ± 31 MBq	51.5	Qualitative, semiquantitative (SUVmax)
Scharitzer M [20]	[^68^Ga]Ga-DOTA-FAPI-04	PET/MRI	3.0-T Biograph mMR, Siemens	167 (149–190) MBq	60	Qualitative, semiquantitative (SUVmax)
Dabir M [21]	[^68^Ga]Ga-DOTA-FAPI-04 and [^68^Ga]Ga-DOTA-FAPI-02	PET/CT	Biograph mCT Flow Siemens	118–340 MBq	60	Qualitative, semiquantitative (SUVmean, SUVmax, TBR)
Qin C [22]	[^68^Ga]Ga-DOTA-FAPI-04	PET/CT and PET/MR	3.0 T, SIGNA TOF-PET/MR, GE Discovery VCT, GE	1.85–3.7 MBq/kg	20–60	Qualitative, semiquantitative (SUVmax)
Röhrich M [23]	[^68^Ga]Ga-DOTA-FAPI-46	PET/CT	Biograph mCT Flow Siemens	150–250 MBq	10 *, 60, 180	Qualitative, semiquantitative (SUVmax)
Sviridenko A [24]	[^68^Ga]Ga-DOTA-FAPI-46	PET/CT	Discovery DMI; GE	220 ± 34 MBq	30, 60, 120	Qualitative, semiquantitative (SUVmax, TBR)
Wang X [25]	[^68^Ga]Ga-DOTA-FAPI-04	PET/CT	Polestar m660, SinoUnion	107.4 ± 26.5 MBq	60	Qualitative, semiquantitative (SUVmax, SUVmean, MV)
Bergmann C [26]	[^68^Ga]Ga-DOTA-FAPI-04	PET/CT	Biograph mCT Flow Siemens	150–250 MBq	60	Qualitative, semiquantitative (SUVmax, SUVmean, MAV, TL-FAPI)
Finke D [27]	[^68^Ga]GA-FAPI	PET/CT	Biograph mCT FlowTM Siemens	122–336 MBq	60	Qualitative, semiquantitative (SUVmax)
Liu H [28]	[^68^Ga]Ga-DOTA-FAPI-04	PET/CT	United Imaging UMI 780	1.85–2.59 MBq/kg	40–60	Qualitative, semiquantitative (SUVmax, TBR)
Luo Y [29]	[^68^Ga]Ga-DOTA-FAPI-04	PET/CT	Biograph 64 TruePoint TrueV [Siemens] or Polestar m660	55.5–162.8 MBq	54.4	Qualitative, semiquantitative (SUVmax)
Zheng S [30]	[^68^Ga]Ga-DOTA-FAPI-04	PET/CT	Biograph mCT 64, Siemens	3.7 MBq/kg	30–60	Qualitative, semiquantitative (SUVmax)
Zhou Y [31]	[^68^Ga]Ga-DOTA-FAPI-04	PET/CT	Na	1.85–2.59 MBq/kg	50–60	Qualitative, semiquantitative (SUVmean liver, SUVmax kidney, TBR)
Schmidkonz C [32]	[^68^Ga]Ga-DOTA-FAPI-04	PET/CT	Biograph mCT 40, Siemens	Na	Na	Qualitative, semiquantitative (SUVmax, SUVmean, MAV, TL-FAPI)

Na: not available; PET/CT: positron emission tomography/computed tomography; MRI: magnetic resonance imaging; MAV: metabolic active volume, TL-FAPI: total lesion FAPI; TBR: target to background; MV: molecular volume; FAV: fibrotic active volume. * In three patients, dynamic scans over 40 min instead of scans after 10 min were acquired.

## Data Availability

Data sharing is not applicable.

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
