# Peer review of "The Role of Fibroblast Activation Protein Inhibitor Positron Emission Tomography in Inflammatory and Infectious Diseases: An Updated Systematic Review"

_pharmaceuticals, 2024, doi:10.3390/ph17060716_

Round 1

Reviewer 1 Report

Comments and Suggestions for Authors

This article provides a comprehensive overview of the applications of FAPI PET in the diagnosis of various diseases, including lung interstitial disease, bone and joint diseases, IgG4-related disease and Crohn’s disease. Through meticulous literature search and screening, the paper elaborates on the potential and advantages of FAPI PET in different diagnostic fields. Furthermore, it delved into potential mechanisms of FAPI uptake, indicating that chronic inflammation and tissue remodeling may be key factors affecting FAPI uptake. The paper highlights the importance of prioritizing standardization and consistency in research to ensure a more accurate assessment of the diagnostic performance of FAPI PET. However, these following issues still need to be addressed.

1. The key words of the article can be simplified. The most important key word should be FAPI; PET; inflammation infection and hybrid imaging;

2. The content of the figures is unclear. Figures 1 on page four and Figure 2 on page 13 are indistinct, making it difficult for readers to accurately comprehend the information conveyed by the charts. To enhance the readability of the article, it is recommended that the author review and optimize the clarity of these legends;

3. There is a need for better integration in explaining possible mechanisms. For instance, the passage "Previous studies showed that the metabolism of inflammatory..." in line 362 is linked to the possible mechanism of FAPI uptake in the subsequent paragraph (line 366). It is recommended that the author consolidates these sections for a more cohesive argument.

4. In referring to "radiolabeled white blood cell scintigraphy" as the "best nuclear medicine imaging modality" at line 432, the author should provide more details on the similarities and distinctions between this imaging technique and FAPI PET. Exploring possible avenues for comparison or reasons for the omission of a comparison will assist readers in understanding the role and value of FAPI PET in nuclear medicine imaging.

Author Response

  1. The key words of the article can be simplified. The most important key word should be FAPI; PET; inflammation infection and hybrid imaging;

Reply: Thanks for this suggestion. We modified the keywords according to your suggestions. 

  1. The content of the figures is unclear. Figures 1 on page four and Figure 2 on page 13 are indistinct, making it difficult for readers to accurately comprehend the information conveyed by the charts. To enhance the readability of the article, it is recommended that the author review and optimize the clarity of these legends;

Reply: we enlarged Figure 1 to increase its readability. About Figure 2, it is near to the corresponding text (paragraph 3.4). We have added a phrase in the text and in the legend to better explain Figure 2.

  1. There is a need for better integration in explaining possible mechanisms. For instance, the passage "Previous studies showed that the metabolism of inflammatory..." in line 362 is linked to the possible mechanism of FAPI uptake in the subsequent paragraph (line 366). It is recommended that the author consolidates these sections for a more cohesive argument.

Reply: Thank you for this comment. We have rephrased this part as suggested

  1. In referring to "radiolabeled white blood cell scintigraphy" as the "best nuclear medicine imaging modality" at line 432, the author should provide more details on the similarities and distinctions between this imaging technique and FAPI PET. Exploring possible avenues for comparison or reasons for the omission of a comparison will assist readers in understanding the role and value of FAPI PET in nuclear medicine imaging.

Reply: We understand your point of view. Radiolabeled WBC is a gold standard technique for specific indications, such as septic periprosthetic joint infection, osteomyelitis, vascular graft infections. Unfortunately, direct comparison between WBC scintigraphy and FAPI PET are not present in the literature but from a theoretical point of view these two tools could be complementary and not alternative. We have discussed this opinion in the final part of the discussion.

Reviewer 2 Report

Comments and Suggestions for Authors

The review by Treglia and coworkers is a literature search of papers that address the use of FAP-binding tracers for PET imaging of fibrotic and inflammatory diseases.

The manuscript is well written and provides the reader with a good background on the basis of which the literature search was conducted. The scope of the review is important as it addresses a timely topic in the field of nuclear medicine.

There are only a few aspects and minor concerns that the authors should address when revising their manuscript:

1.       Line 78-87, introduction: In these lines, starting with “inflammatory and infectious diseases, due to the radiolabeled FAPI uptake as a consequence of fibroblast activation in tissues remodeling, e.g. the application of FAPI PET in rheumatic diseases. […]”  (“tissues” = tissue), the authors do not provide important citations, to give the reader a reasonable impression that the field of application of FAPI has been extended to fibrotic diseases in addition to oncology. It would be valuable for the reader to see the pioneering work being cited on this occasion, as was done for the field of oncology.

2.       Line 113:  The date of inclusion of hits in the database was 31-Dec-2023.  However, this review article was submitted on 16-May-2024.  In such a rapidly evolving field, an attempt should be made to submit the review article after a more recent update according to current publications, a point that should be considered during the revision process.

3.       Line 158:  “different kinds of infectious and inflammatory diseases.”  (“different kinds” = various types)  As there is a major discussion in the field as to whether inflammation always induces fibrosis or whether there are mechanisms of independent progression of fibrosis, it might be beneficial to name fibrotic disease states separately and not include them in the field of “inflammation”.

4.       Table 1:   The heading of Table 1 should include a direct reference to the Table2, which would be useful for the reader to see more easily what has been done with the same tracer.

5.       Line 240-270:  Is there any information in the cited papers on the use of histologic evidence of fibrosis versus a marker of inflammation?  Perhaps some judgment should be made in the text as to whether such studies define FDG as an inflammatory agent and FAPI as a marker of fibrosis - or whether there is a need for further studies with histologic redefinition.

6.        Line 306. “However, more solid data is shareable to confirm or controvert these results in this disease or other fields.”  In this context, it should be mentioned that the mechanism of fibroblast differentiation into distinct subsets has been elucidated in detail and correlated with FAPI tracer uptake in a study published in Nature Immunol. (Nat Immunol 25, 682–692 (2024). https://doi.org/10.1038/s41590-024-01774-4), confirming the results.

7.       Line 335. “FAPI was generally considered best in fibrotic inflammatory disorders”: Is the term “fibrotic inflammatory disorders” really defined?  There may be an emerging discussion to redefine what comes first: inflammation or fibroblast differentiation?  With the use of FAPI tracers for in vivo imaging, such research to elucidate disease progression and underlying mechanisms may be achievable - and could be brought to the forefront.

8.       Line 370: “FAP is overexpressed on CAFs’ cell membrane and stroma and can be an indirect expression of extracellular matrix remodeling, typical for example of several inflammatory and infectious conditions.” The authors refer to cancer-associated fibroblasts, although in inflammatory processes TGF-beta is the driving factor, e.g. secreted by synovial activated fibroblasts (SAF), not CAF. The manuscript might benefit from referring to specific mechanisms that activate fibroblasts in fibrotic conditions rather than cancer.

9.       Line 384: “macrophages may express FAP receptors on their cell surface.”  A literature citation is missing to underline this aspect – there is literature precedent related to heart injury.

10.   Line 399:  “e al”  =  et al

11.   Line 430-443:  The lack of large multi-center clinical trials with one FAPI tracer in inflammatory diseases could be mentioned.  Could be recommended for initiation.

Line 510-515: References: highlighted links should be removed.

Comments on the Quality of English Language

Minor editing of English language should be done (see examples of point 1 and 3 in Comments for Authors. 

Author Response

  1. Line 78-87, introduction: In these lines, starting with “inflammatory and infectious diseases, due to the radiolabeled FAPI uptake as a consequence of fibroblast activation in tissues remodeling, e.g. the application of FAPI PET in rheumatic diseases. […]”  (“tissues” = tissue), the authors do not provide important citations, to give the reader a reasonable impression that the field of application of FAPI has been extended to fibrotic diseases in addition to oncology. It would be valuable for the reader to see the pioneering work being cited on this occasion, as was done for the field of oncology.

Reply: We have added a reference and changed the part.

  1. Line 113:  The date of inclusion of hits in the database was 31-Dec-2023.  However, this review article was submitted on 16-May-2024.  In such a rapidly evolving field, an attempt should be made to submit the review article after a more recent update according to current publications, a point that should be considered during the revision process.

Reply: dear reviewer, as you know the process of writing a systematic review is very complex and long. The literature search, the data extraction, the risk of bias assessment need an accurate work and a careful attention. Moreover, the subsequent process of writing and review of all authors cost some time. This is the reason of the delay between December 2023 and the submission of the manuscript. Five months form the last search to the manuscript submission are usually considered acceptable for an updatd review. The idea to update the work with a new deadline does not seems feasible at this point as this means to repeat all the process modifying all the results.

  1. Line 158:  “different kinds of infectious and inflammatory diseases.”  (“different kinds” = various types)  As there is a major discussion in the field as to whether inflammation always induces fibrosis or whether there are mechanisms of independent progression of fibrosis, it might be beneficial to name fibrotic disease states separately and not include them in the field of “inflammation”.

Reply: We agree with your comments. Fibrosis can be a mechanism directly related with inflammatory diseases (it is not perfectly clear if as a consequence of as a previous moment). However, the aim of this review is to analyzed the role of FAPI PET in infectious and inflammatory diseases and our research showed that the studied that evaluated these diseases demonstrated a role of FAPI in the evaluation of fibrotic tissues. This was a consequence of our investigation, so it was not supposed before starting the research. A manuscript focused specifically on fibrotic diseases should be desirable.

  1. Table 1:   The heading of Table 1 should include a direct reference to the Table2, which would be useful for the reader to see more easily what has been done with the same tracer.

Reply: Table 1 and 2 contain different types of data on the same manuscripts. We prefer to have more than one table in order to show more information.

  1. Line 240-270:  Is there any information in the cited papers on the use of histologic evidence of fibrosis versus a marker of inflammation?  Perhaps some judgment should be made in the text as to whether such studies define FDG as an inflammatory agent and FAPI as a marker of fibrosis - or whether there is a need for further studies with histologic redefinition.

Reply: Good question. Unfortunately, the evidence now are not so clear and the open questions remain. The need of further studies with the aim to define from an histological point of view is brilliant and correct.

  1. Line 306. “However, more solid data is shareable to confirm or controvert these results in this disease or other fields.” In this context, it should be mentioned that the mechanism of fibroblast differentiation into distinct subsets has been elucidated in detail and correlated with FAPI tracer uptake in a study published in Nature Immunol. (Nat Immunol 25, 682–692 (2024). https://doi.org/10.1038/s41590-024-01774-4), confirming the results.

Reply: You suggested reference is right, but as answered before, our review finished on 31-Dec-2023, while this paper was published on 2024 February 23. So at the time of the writing we did not have the possibility to read this interesting work, as this was not included in the literature search.

  1. Line 335. “FAPI was generally considered best in fibrotic inflammatory disorders”: Is the term “fibrotic inflammatory disorders” really defined? There may be an emerging discussion to redefine what comes first: inflammation or fibroblast differentiation? With the use of FAPI tracers for in vivo imaging, such research to elucidate disease progression and underlying mechanisms may be achievable - and could be brought to the forefront.

Reply: Very interesting and appropriate observation. We have modified the discussion following this point.

  1. Line 370: “FAP is overexpressed on CAFs’ cell membrane and stroma and can be an indirect expression of extracellular matrix remodeling, typical for example of several inflammatory and infectious conditions.” The authors refer to cancer-associated fibroblasts, although in inflammatory processes TGF-beta is the driving factor, e.g. secreted by synovial activated fibroblasts (SAF), not CAF. The manuscript might benefit from referring to specific mechanisms that activate fibroblasts in fibrotic conditions rather than cancer.

Reply: Very interesting and appropriate observation. We have modified the discussion following this point.

  1. Line 384: “macrophages may express FAP receptors on their cell surface.”  A literature citation is missing to underline this aspect – there is literature precedent related to heart injury.

Reply: Good point. We have added this reference “Buechler MB, Fu W, Turley SJ. Fibroblast-macrophage reciprocal interactions in health, fibrosis, and cancer. Immunity. 2021 May 11;54(5):903-915. doi: 10.1016/j.immuni.2021.04.021. PMID: 33979587”.

  1. Line 399:  “e al”  =  et al

Reply: Sorry for this error, we corrected it

  1. Line 430-443:  The lack of large multi-center clinical trials with one FAPI tracer in inflammatory diseases could be mentioned.  Could be recommended for initiation.

Reply: Good observation, We have added this observation in the final part of the discussion

Line 510-515: References: highlighted links should be removed.

Reply: Sorry for this mistake , we have corrected it